# On the Directivity of Lamb Waves Generated by Wedge PZT Actuator in Thin CFRP Panel

**DOI:** 10.3390/ma13040907

**Published:** 2020-02-18

**Authors:** Sergey Shevtsov, Valery Chebanenko, Maria Shevtsova, Shun-Hsyung Chang, Evgenia Kirillova, Evgeny Rozhkov

**Affiliations:** 1Aircraft Engineering Dept, EFRC The Southern Scientific Centre of Russian Academy of Science, 344006 Rostov-on-Don, Russia; valera@chebanenko.ru (V.C.); mariamarcs@bk.ru (M.S.); 2Microelectronic Dept, the National Kaohsiung University of Science and Technology, Kaohsiung 81157, Taiwan; stephenshchang@me.com; 3Faculty of Architecture and Civil Engineering, RheinMain University of Applied Science, 65197 Wiesbaden, Germany; kirillova@web.de; 4Institute of Mechanics and Applied Mathematics, Southern Federal University, 344006 Rostov-on-Don, Russia; erozhkov@sfedu.ru

**Keywords:** acoustic based SHM, orthotropic polymeric composites, Lamb waves, horizontally polarized SH waves, angle-beam wedge transducer, waves directivity

## Abstract

This paper addresses investigation of guided-wave excitation by angle-beam wedge piezoelectric (PZT) transducers in multilayered composite plate structure with orthotropic symmetry of the material. The aim of the present study is to determine the capability of such actuators to provide the controlled generation of an acoustic wave of a desirable type with the necessary wavelength, propagation distance and directivity. The studied CFRP (Carbon Fiber Reinforced Plastic) panel is considered to be homogenous, with effective elastic moduli and anisotropic structural damping, whose parameters were determined experimentally. According to the results of dispersion analysis and taking into account the data of wave attenuation in a highly damping CFRP composite, the two types of propagating waves A0 and S0 were considered theoretically and experimentally in the frequency range of 10–100 kHz. Using the results of a previous study, we reconstructed the structure of the wedge actuator, to develop its finite-element (FE) model, and a modal analysis was carried out that revealed the most intense natural vibration modes and their eigenfrequencies within the frequency range used. Both experimental and numerical studies of the generation, propagation, directivity and attenuation of waves in the orthotropic composite panel under study revealed the influence of the angular orientation of the actuator on the formation of wave patterns and allowed to determine the capabilities of the wave’s directivity control.

## 1. Introduction

The wide spread of the glass-fiber (GFRP) and carbon-fiber reinforced polymeric composites in aircraft, automotive and shipbuilding industries requires a reliable, cost-effective and easy-to-use means for the Structural Health Monitoring (SHM) and Nondestructive Evaluation (NDE) to supply the quality and defect-free products. Such methods and the necessary tools are most in demand in manufacturing of load carrying structures, whose defects can cause emergencies or some dangerous failures. Among the known SHM and NDE methods of the polymeric composite materials the acoustic based methods are most effective and used in practice [1,2,3,4]. The common difficulties at the design of efficient acoustic based SHM of polymeric composites are the structural anisotropy of the elastic [2] and high damping properties [1,5,6,7,8,9] of the materials that reduces the wave propagation distance [9,10]. The carbon-fiber reinforced composite materials present additional challenges for inspection due to their conductivity of the fibers, the insulating properties of the matrix and the fact that a damage often occurs beneath the visible surface [2]. Meanwhile, the collective skills of many researchers confirm that, in some cases, the Lamb-wave-based SHM techniques can provide more reliable information about the damage presence and the severity than another method, and provide the possibility of determining a damage location due to their local response nature. These imperfections that can be detected by the acoustic SHM are the delaminations, the inclusions, the uneven resin cure areas and the porosity [11], the dry spots, which can arise during liquid composite molding [12,13], and even the errors of lay-up stacking orientation [14].

In order to detect, identify the geometric location and the severity of the probable imperfection, a proper choice of the wave type, length and intensity within the whole scanned area should be surely fulfilled. That is the first task of each acoustic-based SHM technique [15,16]. This choice should provide the necessary resolution, wave propagation distance and directivity, which depend on the composite’s elastic, and damping properties anisotropy, on the excited wave type and the frequency.

In one of the early works [17] both the static and the dynamic values of the storage moduli of the laminated beams were predicted from unidirectionally reinforced ply data, and their values were used to predict the wave speed in two quasi-isotropic laminates. The authors of the article [16] proposed a new approach, which is based on the assembling the laminate global matrix from the individual layer matrices, yields traveling waves along both directions in the plane of the plate. The developed approach was applied to the laminates with different lay-ups. The quasi-isotropic laminates were studied experimentally in [14] for the detection of the lay-up stacking orientation. The paper [8], which experimentally studied the Lamb wave attenuation in the quasi-isotropic plastic, demonstrated a sufficient decrease of the propagating wave amplitude with angle between the main axis of anisotropy and wave path. Hu with coworkers [11] investigated the A0 and S0 waves propagation and their sensitivity to the delamination in cross-ply laminates. It should be noted that all of the aforementioned works [2,8,11,14], as well as the other articles [16,17,18] that investigated the influence of the anisotropy of the elastic properties of the material on the propagation of acoustic waves, tested the proposed models and calculation methods on quasi-isotropic and cross-ply laminates.

Another important property characterizing the ability of acoustic waves to interact with possible defects is the range of their propagation, or their opposite property is attenuation. The authors of [9,10] divide the phenomena, which determine the distance of the wave propagation, into four kinds:-Geometric spreading;-Material damping;-Wave dispersion;-Dissipation into adjacent media.

The papers [10,19] attributed to the main reasons for the material damping the following phenomena:-Viscoelastic nature of matrix and/or fiber materials;-Damping due to interphase interaction;-Damping due to damage;-Viscoplastic damping due to the presence of high stress and strain concentration that exists in local regions between fibers;-Thermo-elastic damping due to the cyclic heat flow from the region of compressive stress to the region of the tensile stress.

This list makes obvious the damping’s anisotropy of the multilayered composites with differently oriented layers. The propagation of attenuating waves generated by the omnidirectional circular PZT actuators in a CFRP plate with significant anisotropy of elastic and dissipative properties was studied in the authors’ works [20,21], where the properties of the orthotropic material were previously determined as effective modules in accordance with the methodology considered in [22]. But the dissipative properties of the fiber reinforced composites depend on the many factors, and these dependencies should be taken into account at the estimation of the range of wave advancement to supply the desired parameters of SHM system. These dependencies have been studied by many researchers.

In the early study [23], it was established that the fiber material usually has extremely low damping capacities, and so it can be considered not to contribute to the damping of the composite, whereas a large proportion of the dissipated energy must be in the matrix. The work [24] reported about many experiments implemented with different kinds of CFRP, and GFRP laminas demonstrated a significant growth of the damping parameters (up to ten times and more) with the change of a cyclic strain orientation relative to the fibers’ direction. Moreover, these damping parameters are frequency dependent, and they always increase with the frequency, especially for the cyclic strain normal to the fibers’ direction. This work confirmed experimentally the validity of the Hashin equation [25], which describes the dependence of the flexural loss factor, *η*, on the fiber volume fraction in composites on the loss factor of the matrix, and on the real parts of the complex modulus of the fiber and resin, respectively. The theoretical approach for determination of the frequency dependent loss factor of the laminated composites, which are proposed in the report [7], is based on the similar dependencies of the lamina’s loss factors at the longitudinal, transversal and shear cyclic strains, and on the classical lamination theory equations [26].

The basic elasticity equation for unidirectional composites, together with the Adams–Bacon damping criterion, were utilized in [27] for the prediction of the moduli and a flexural damping of anisotropic CFRP and GFRP beams with respect to fiber orientation. This criterion postulated that the energy dissipation in a thin unidirectional lamina is the sum of separable energy dissipations due to *σ_x_*, *σ_y_*, and *τ_xy_*. The proposed approach was used to obtain the variation of elastic moduli and the damping capacities of the beams with the different angle-ply orientations. The calculation results, which were confirmed experimentally, revealed very intensive angular dependencies of the damping parameters for the orthotropic composites. Moreover, the structural damping increased with the increase of the deviation angle from the main axis of anisotropy [27].

Validity of these conclusions were confirmed in the article [18], which contains the results obtained at the experimental study of the wave attenuation along and across unidirectional CFRP, cross-ply and quasi-isotropic composites for the different kinds of acoustic waves propagations at the varied frequencies. The similar results were presented in the later paper [19].

In order to estimate the wave propagation distance at the implementation of the acoustic-based SHM systems for the cross-ply and the quasi-isotropic composites, the analogous studies were carried out, and the corresponding results were presented in the works [5,8]. The present results demonstrated a sufficient decrease of the propagating Lamb wave amplitude with the angle between main axis of anisotropy and wave path. In addition, it was established that the attenuation of the flexural A0 mode is much higher than the extensional S0 mode. Another important result reveals that quasi-isotropic [0°, 45–45°, 90°]_2s_ plates are useful in design, but the stress wave inspection is difficult due to high attenuation of both Lamb modes. It is obvious that this conclusion relates to all CFRP and GFRP plates with complex lay-up and anisotropy. The authors of the paper [8] concluded that knowledge of the attenuation and directivity of the waves, depending on the anisotropy of the elastic and damping properties of the material for specific types of waves and their excitation frequencies, will benefit future acoustic based SHM and NDE applications in composite material design and technology.

The difficulties of the reliable estimation of the waves’ attenuation in the multilayered composites by using some model of the wave attenuation led to the development of the experimental technique [9] for the determination of the attenuation properties directly at the study of the excited waves propagation. The comparative analysis of Equation (1) with the experimental results that were performed in [9] demonstrated that a reduction in the amplitude of a Lamb wave is very steep nearer to the excitation source, and this reduction in amplitude is independent of the material attenuation. When moved away from the source, the reduction in amplitude is due to both geometry and material damping, but the latter dominates. The experimental technique [9], which allows us to determine the angular distribution of the attenuation coefficient, was used in our previous work [19]. These results have been used in the present paper.

Despite the fact that many theoretical papers report frequencies of the used waves of the order of several MHz, articles presenting the results of neatly performed experiments inform of much lower frequencies. Schultz and Tsai [17] reported that they were able to reliably measure the wave attenuation in the laminated beams at the frequencies no more than 10 kHz. The later studies [1,9,10,11] using modern advanced equipment were able to operate at higher frequencies. The results of the acoustic-based damage detection presented in [1] were obtained at the relatively low excitation frequency, in the range of 15–50 kHz, that is due to a very high Lamb wave attenuation in the studied composite panels. The work described in [11], which compared the sensitivity of A0 and S0 Lamb wave to the delamination in CFRP cross-ply laminated beams, established that, due to the very intensive Lamb wave attenuation, the two excitation frequencies, 45 KHz and 80 KHz, were used for the diagnosis of the delamination. The authors of this paper securely recorded an attenuation of the Lamb waves excited by the circumferential PZT actuators in the orthotropic CFRP panel, using frequencies within the range of 10–100 kHz [20]. Apparently, the boundaries of this range determine the frequencies at which it is possible to provide a sufficient propagation distance for the traveling Lamb waves.

The fact that the attenuation of the waves in the anisotropic materials is also anisotropic makes us pay special attention to the possibility of controlling the directivity of the generated waves. The abilities of the sources of acoustic waves, including their frequency ranges, intensity and directivity, are given in the comprehensive review paper [3]. Raghavan and Cesnik, in their paper [16], noted that there is little or no theoretical basis provided by the researchers for the optimal choice of the various testing parameters involved, such as transducer geometry, dimensions, location and materials, excitation frequency, and bandwidth, especially, for the anisotropic tested material. Using a modified semi-analytical approach based on the spatial Fourier inversion and the simplest delta-like and Heaviside forcing functions due to the piezoelectric actuators of circular, ring-shaped and rectangular shapes, which act on the plate surface by the shear stress, and neglecting by their enough complex, frequency dependent distribution on interface, they determined the dependence of generated Lamb waves on the actuators’ properties. The developed approach was successfully applied to the two quasi-isotropic symmetric laminates with different lay-ups. Such an approach, which can be used for the omnidirectional transducer with sufficient simplification of their forcing functions, has been used by many researchers. Its advantages and limitations were analyzed in the experimental work [28]. In this work, a scanning laser Doppler vibrometer (SLDV) was used as a receiver with the ability of the spatially dense sensing, allowing us to analyze the wave propagating at the different angles from the omnidirectional wave exciter. Ostiguy with co-workers argued in their paper [29] that, at the analytical modeling of guided wave generation by a circular PZT and propagation, the analytical formulations need to consider (1) the dependency of phase velocity and damping as a function of angle, (2) the steering effect on guided wave propagation caused by the anisotropy of the structure, and (3) the full transducer dynamics. The most important feature of the performed study was the dynamic analysis of the system “circular PZT actuator-composite panel”. It allowed us to reveal the in-plane stress distribution on the interface and to obtain the qualitative estimation of the influence of the anisotropic damping on the wave attenuation. The radial shear stress calculated at specific frequencies showed that the in-plane shear stress (1) varies in amplitude over the frequency range of interest, (2) were dependent on the orientation and (3) were not only located at the edge of the PZT. The finite-element implementation of the similar approach was applied by the authors of the present paper at the study of the directivity of Lamb waves generated by the circular PZT actuators of different dimensions at the varied excitation frequency [21]. In the reviewed works, an important limitation inherent to omnidirectional actuators was revealed. This limitation is an impossibility to controllable change of their directivity. To overcome it, the following approaches were developed, applying to the problem of controlled directivity of the excited guided waves: the use of the distributed PZT transducers array, which generates selectively the desirable Lamb modes [30]; active fiber composites and macro-fiber composites due to their broad frequency of operation, high reliability and easy installation [6,15]; the interdigital transducers, which are able to generate bidirectional waves with relatively low side lobe level [6]. However, SHM of the large composite structures, where the acoustic waves generated by the spatially fixed exciter quickly attenuate and cannot spread over long distance, is difficult. This makes it necessary to use actuators that make it easy to change their location and angular orientation on the test structure that is very important for the anisotropic materials.

Among the many modern and widespread designs of the transducer for the acoustic waves’ generation, the angle-beam wedge transducers, which are traditionally used for the SHM of thick-walled structures made of isotropic metallic materials, were considered in the survey [3]. The first fundamental study of the wedge transducers was presented in the monograph [31], where the conditions for the optimal transformation of the longitudinal waves generated by the PZT plate to the surface Rayleigh wave in the tested structure.

Some modernizations of the wedge actuators allowing to eliminate the reflections at the wedge-load interface and multiple reflections inside the wedge were described in [32,33,34,35,36]. The work [34], which assumed a very simplified roughly parabolic pressure distribution on the interface between the tested panel and angle-beam actuator, reported that the width of the so-called “phase velocity spectrum” is dependent only on the ratio of loading length to wavelength of the mode being generated. It was proven that, for a given frequency, there will be a finite number of real wavenumbers (that can be excited), satisfying the dispersion equation at the general case of anisotropic tested materials.

The paper [35] studied the effect of dimensions, the shape and the aperture on the frequency response and the directivity of the waves generated by the angle-beam wedge transducer in an isotropic half-space, using the boundary element method. The authors presented the results of a comprehensive parametric study in an effort to establish the guidelines and the criteria for the optimum wedge transducer design. The main objective of the study was the optimal choice of the wedge shape and dimensions to avoid spurious eigenmodes or unexpected wave propagations within the wedge. In the paper [36], which studies the similar problems, the effect of the shape and size of wedge-shaped substrates on the whole transducer system was discussed, and the design of a transducer was optimized to supply its performance when it generates the acoustic wave in a tested thick-walled structure made of isotropic material by using a 2D finite-element (FE) model.

The later studies of the wedge actuators excited the acoustic waves in an isotropic elastic half-space by using more sophisticated three-dimensional (3D) models that rely on highly numerical approaches such as finite elements, boundary elements, multi-Gaussian beam (MGB) or the distributed point source (PSM) methods [37,38,39,40].

To find out the possibility of generating the directed acoustic waves in thin-walled structures using a wedge actuator, the authors of the current article investigated this problem by reconstructing the dynamics of the entire system, including the structure of the actuator and the quasi-isotropic plastic plate under test. To do this, a 3D coupled transient problem of piezoelectricity and mechanics for a system of bodies made of elastic materials having different structural damping was formulated and solved by the finite-element method. The results of our numerical investigations, which were confirmed experimentally, revealed that upper limit of the excited frequency range was determined by the damping of the material of structure under testing. The highest efficiency, intensity and propagation distance of the waves generated by the wedge actuator were achieved when the waves were excited at the eigenfrequencies of its natural vibration modes. The directivity of the propagated waves depended on the shear stress distribution on the actuator-panel interface, i.e., on the selected actuator’s oscillation mode.

This paper is aimed to reveal the possibility of generating directed acoustic waves by the angle-beam wedge actuator in thin-walled structures made of highly anisotropic material. As objects of our study we used the orthotropic CFRP panel and the angle-beam wedge actuator Olympus V414-SB-ABWS (Waltham, MA, USA), which has been modeled and experimentally investigated in our works [41]. By using experimentally verified frequency response functions for the displacement amplitudes and the electric current consumption of the actuator, the testing frequencies were selected. The directivity of the excited anti-symmetric A0 Lamb waves was studied at the different orientation of actuator relative to the main axis of orthotropic material. These directivity diagrams are presented at the end of this paper, together with the radial shear stress distribution on interface, to demonstrate the interdependence of these stress and the directivity diagrams. All obtained results and the developed technique are intended to clarify the abilities of the angle-beam wedge actuator at its use for SHM of orthotropic composite materials.

## 2. Materials and Methods

Our investigation included three stages, and besides, each stage was performed both experimentally and theoretically. First, the dispersion equations to find the kinds and wavelengths of the acoustic waves that can be excited in the studied composite structure were determined. In order to solve the dispersion equations all mechanical properties for the studied composite panel made of orthotropic CFRP are adopted from our early research [20,21].

To simulate the wave propagation in the studied CFRP panel, the numerical finite-element (FE) model of the used wedge transducer was identified on the base of the results of its experimental investigation, which provided data on the dynamic properties of the actuator required to develop an adequate device’s model.

The full-scale experiment and FE simulation of the acoustic wave propagation in the orthotropic panel were implemented at the chosen excitation frequencies and variation of the wedge actuator orientation relative to the main axis of orthotropic material of CFRP panel. The results of these studies are presented in the form of the wave directivity diagrams and angular dependencies of the propagated wave attenuation. The obtained results matched with the distributions of the stresses and displacements on the interface between actuator’s footprint and the tested panel to understand the ability of the wedge transducer to generate the spatially oriented wave beams.

### 2.1. Dispersion Analysis of the Waves That Can Be Excited in CFRP Panel under Study

The investigated CFRP panel was manufactured by the laying-up of unidirectional carbon-fiber epoxy based prepreg KMKY-2m.120 with mechanical properties $E_{long}=100\pm 12$ GPa, $E_{trans}=8\pm 2$ GPa according to the scheme [0°; −30°; 30°; 90°]_2s_ to produce a symmetric balanced laminate. Its elastic properties were determined by two independent methods. The longitudinal *E*_1_, transversal *E*_2_, in-plane shear *G*_12_ moduli, and Poisson ratio *ν*_12_ were calculated, using prepreg’s manufacturer data of lamina elastic properties and equations of the classical lamination theory [26]. In order to obtain more realistic values of the elastic constants of the ready composite, the experimental technique described in our paper [19] was used. This technique included the determination of the effective moduli experimentally, according to standards ASTM D 3039-95 [42], D 5379-93 [43], D 2344-89 [44], and a refinement technique to reduce the effect of non-ideal experimental conditions [21]. The through thickness module *E*_3_, the interlaminar shear moduli *G*_23_, *G*_31_, and the Poisson ratios *ν*_23_ and *ν*_31_, unavailable for reliable experimental measurement, were calculated by using FE simulation of the static tests for the multilayered composite [45]. The confidence intervals for all elastic constants were calculated, using the experimentally measured ones, obtained for every 5 tested specimens. The final results of the engineering constants of the studied CFRP panel, which are presented in Table 1, are typical for an orthotropic structural symmetry of composites.

The dispersion analysis of the acoustic-waves phenomena in the studied CFRP panel was performed, using the partial-wave technique in anisotropic media, which assumes that the partial waves (or transverse resonance) have the solutions [10,29,46,47]:(1)$$u=u_{0}\exp\left[i\cdot \left(k_{x}x+k_{z}z-\omega t\right)\right]$$

Substituting Equation (4) into the governing equations of elastic waves in anisotropic media [47] gives the following:(2)$$\left[\begin{array}{ccc} A_{11} & A_{12} & A_{13}\\ A_{12} & A_{22} & A_{23}\\ A_{13} & A_{23} & A_{23} \end{array}\right]\cdot \left[\begin{array}{c} u_{x}^{0}\\ u_{y}^{0}\\ u_{z}^{0} \end{array}\right]=\left[\begin{array}{c} 0\\ 0\\ 0 \end{array}\right],$$
where the matrix coefficients, $A_{m,l}$, are expressed through the elements of the orthotropic material’s stiffness matrix, $C_{i,j}$, the material density, ρ, the wave numbers, $\begin{array}{cc} k_{j} & j=x,y,z \end{array}$, the plate thickness b, on the angle, θ, orientation of the wave path relative to the main orthotropy axis, x, and on the angular velocity, ω [47]. To find the eigenvalues of Equation (2), the zero-determinant condition was imposed:(3)[A]=0.

For the Lamb waves, made up of partial waves polarized in the sagittal plane, the following dispersion expressions [47] was used:(4)$$Q_{1}\tan^{\pm 1}\left(\frac{\pi }{2}\sqrt{R_{2}}\right)=Q_{2}\tan^{\pm 1}\left(\frac{\pi }{2}\sqrt{R_{3}}\right),$$
where $Q_{1,2}$ are expressed through the three eigenvalues, $\begin{array}{cc} R_{m}, & m=1,2,3 \end{array}$ for $k_{z}^{2}$, of Equation (3). These eigenvalues depend on the elements of the A matrix [47]. In Equation (4), the “−” exponent corresponds to the symmetric solutions, and the “+” exponent to the antisymmetric solutions. In order to solve this equation, the set of the engineering constants in the Table 1 was transformed to the stiffness matrix, C, and the left part of the transcendental Equation (4) was tabulated at the varying frequency in a range 10–100 kHz, to find the zero values of this determinant. Our calculation results for the chosen frequency range confirmed existence of only one antisymmetric solution, A0. The results in the form of the dispersion curves for two values of angle $\begin{array}{ccc} \theta =0{}^{\circ} & and & \theta =90{}^{\circ} \end{array}$ are shown in Figure 1a,b, together with the experimentally measured values at the frequencies, which correspond to the natural oscillation modes of actuator (see below).

If the wave propagation direction is in the *x* or *y* direction, the resulting dispersion equations for SH waves, which propagate along these directions, can be presented in the simplest form:(5)$$\begin{array}{cccc} \frac{{\left(2bf\right)}^{2}}{C_{4,4}}\left(\rho -\frac{C_{6,6}}{c^{2}}\right)=n^{2} & at & \theta =0{}^{\circ}; & \begin{array}{ccc} \frac{{\left(2bf\right)}^{2}}{C_{5,5}}\left(\rho -\frac{C_{6,6}}{c^{2}}\right)=n^{2} & at & \theta =90{}^{\circ} \end{array} \end{array},$$
where c is a speed of the SH wave propagation, and n is a non-negative integer. Both of these equations have the real solution only at the n=0. Hence, the wave speed for these directions of two non-dispersive SH waves are equal to $c_{SH}=\sqrt{C_{6,6}/\rho }$. The dependencies of the SS0 wave speed and wavelength on the frequency are shown in Figure 2a,b, together with the results of the FE simulation at the actuator’s first eigenfrequencies.

Comparison of Figure 1b and Figure 2b demonstrates that the wavelength of A0 Lamb waves mode is 2–5 time less than the wavelength of SS0 horizontally polarized shear wave, i.e., spatial defects resolution of A0 waves can be significantly better that is fully consistent with the results of the papers [11,20,22,30].

### 2.2. The Finite-Element Model of the Angle-Beam Wedge Actuator, Generating the Wave in a Thin CFRP Panel

The second important component of the studied system was the angle-beam wedge actuator, which excited the acoustic waves in the CFRP panel under consideration. In order to develop the finite-element model of the full system’s dynamics that should be quantitatively comparable with the response of a real-world experimental system, the electrical and mechanical subsystems properties of the modeled actuator need to be as close as possible to the properties of the used Olympus actuator. Identification technique of its structure and the electro-mechanical properties were presented in our paper [41]. The sources, which have been used to adequately build the transducer’s structure and replicate its dynamics in FE model, were the technical notes about the ultrasonic principles important to wedge transducer application and design, including the scheme and main requirements to the active element, backing, matching layers and the sound path presented by Olympus Co.^®^ [48], the outer dimensions of the used actuator (see Figure 3), the frequency response functions for the electric current through PZT active plate, and out-of-plane displacement amplitude on the actuator’s footprint (see Figure 4a,b), which have been determined experimentally. Only averaged normal displacement amplitudes of the actuator’s contact surface were measured experimentally by using the VibroGo single-point vibrometer (Polytec Co.^®^, Irvin, CA, USA), which was sensitive to the normal acceleration at the frequencies up to 100 kHz. The displacement amplitudes were calculated after the double integration of the sensor’s signal over time for each frequency. In order to compare the experimental and the simulation results, the displacements of the modeled actuator contact surface were averaged within the footprint. The tangent displacement could not be measured experimentally by the available means. They are presented in Figure 4c,d as the results of computer simulations of the actuator’s FE model, whose geometry is present in Figure 5. This model was implemented in the Comsol Multiphysics FE software (3.5a, COMSOL AB, Stockholm, Sweden), which contains built-in Structural Mechanics and Piezoelectric computations modules designed to solve classical equations of anisotropic three-dimensional elasticity and the piezoelectricity [26,45,46]. This ability allows it to easily tune elastic, damping properties of each actuator’s component, and to orient the polarization direction of the active PZT plate along the sound path made of Lucite.

These results made it possible to obtain preliminary estimates of the waves excitation frequencies; however, these frequencies were subject to refinement due to the interaction of the dynamic system of the actuator and the studied orthotropic CFRP panel.

### 2.3. Investigation of the Wave Propagation Generated by the Oriented Angle-Beam Wedge Actuator in Orthotropic CFRP Panel: Experimental and Numerical Studies

An analysis of the wave attenuation in a damped plate was based on the Equations (9) and (10):(6)$$\phi \left(r,t\right)=\left(A/\sqrt{r}\right)\cdot e^{-\eta r}e^{i\left(\omega t-\gamma r\right)},$$
where *r* is the distance between the wave source and the amplitude measuring point, *A* is the magnitude of the signal, *ϕ*(*r*,*t*) is a generic disturbance that propagates in space as a wave, *γ* is the wave number, *ω* is the angular frequency, and *η* is the wave damping coefficient. The experimental study, which was carried out and described in our paper [20], included measurement at the varied frequencies and angular orientations of the Lamb wave amplitudes for the different distances from the omnidirectional circular actuator. Then a numerical processing was performed according to technique proposed in [9] to separate the geometric and damping attenuations. These results were used to identify the Rayleigh damping coefficients of the material, according to the technique described in [8].

Despite the very intensive acoustic wave attenuation, whose angular distribution was measured by the authors and presented in [20] (see Figure 6), the square CFRP panel was equipped with the porous rubbery absorber to fully eliminate the wave reflections from the edge of the plate (see Figure 7a). The FE model of the studied system provided this attenuation inside the circle of radius 27.5 cm by the Rayleigh damping coefficients, which were determined experimentally in [20], depend on the orientation in a polar coordinate system. However, outside this circle, the damping coefficients increased quadratically, while maintaining C^0^ and C^1^ continuities (of the damping coefficients distribution and their spatial derivatives, respectively) at the boundary of the circle. By such a way, the perfectly matched layer was designed to eliminate the wave reflection. On average, for different frequencies, the model consisted of 250–300 thousand of the finite elements; the number of degrees of freedom of the problem was 1.2 × 10^6^–1.5 × 10^6^. The calculation time of one variant with 7–8 fluctuations in the control voltage varied from 4 to 8 h on a computer with an i7 processor and 32 GB of RAM.

In each test, for the variable frequencies and angles of the actuator orientation, the amplitude of the control signal gradually increased, reaching 50 V, and then stabilized (see Figure 8). These actuator driving signals were prepared by using the AWG/AFG software for Windows (V3.4, Tektronix, Inc., Beaverton, OR, USA). They were transmitted to a Tektronix arbitrary function generator and amplified by PA94 piezodrivers (Apex Co., Tucson, AZ, USA). The actuator’s and sensors’ (STEMiNC SMD10T2R111WL, Davenport, FL, USA) signals were registered by oscilloscope LeCroy (Teledyne LeCroy GmBH, Heidelberg, Germany) and stored for each testing condition in the files, which were then numerically processed.

To exclude possible interference of direct Lamb waves and reflected horizontally polarized SH waves, whose damping is much less, the intensity of the wave A0 detected by the sensors was estimated by averaging the absolute value of the sensors signal during three periods of oscillation, when the signal amplitude stabilized.

The directivity of the waves generated by the wedge actuator was studied at three frequencies, 15 kHz, 30 kHz, and 65 kHz, which were chosen due to the highest intensity of the waves excited at these frequencies, which are the eigenfrequencies of the actuator’s structure. The directivity of only anti-symmetric Lamb waves A0 was experimentally investigated. The very long wavelengths (low spatial resolution) and highly distorted wavefront of the SS0 waves at the used frequencies even at the *θ* = 0° (see Figure 9) make the horizontally polarized shear waves unpromising for detection of the minor defects in highly attenuated composites.

The directivity diagrams for the Lamb waves excited by the actuator, whose angular orientation relatively to the main orthotropy axis of the CFRP panel was changed discretely in interval θ∈[0°;90°] with a step 15°, were implemented using two identical sensors 15 cm and 20 cm away from the center of the actuator’s footprint (see Figure 7b). For each actuator’s position, these sensors were registering the signals depending on the out-of-plane displacement at their locations. In order to reduce the experimental noise, the registration of radiation directivity was carried out with an angular step of 15 degrees for two closely spaced distances. Due to the attenuation, the signals recorded at distances of 15 cm and 20 cm were unequal. Therefore, the values of both radiation patterns were normalized and averaged. Their values are presented in Figure 10a–f.

At the numerical simulation, rotation of the actuator direction relative to the main axis of orthotropic anisotropy was carried out by turning its local coordinate system around the axis of the global system, which was normal to the panel plane and passed through its center.

## 3. Results and Discussion

### 3.1. Comparison of the Experimentally Measured and Numerically Calculated A0 Wave Directivities

Some examples of directivity diagrams obtained after the experimental tests and numerical simulations for the different wave excitation frequencies are presented together in Figure 10a–f. These plots demonstrate a radical abrupt change in the main orientation of the waves upon rotation of the actuator, which was confirmed by comparing the calculated data with the experiment. The variability of the wave radiation directivity with frequency is illustrated in Figure 11, which shows data calculated only by FE model for the entire range of angles of the actuator orientation.

These directivity diagrams demonstrate some features that are inherent for highly anisotropic materials. Among these features, which limit the ability of directional wave generation in the composite structures, are the following:-Wave propagation mainly in the direction of greatest structural stiffness and minimal attenuation;-Deviation of the maximum of the radiation directivity in the direction of the main axis of orthotropic anisotropy, which is confirmed by the fact that when the actuator rotates around an axis normal to the panel surface by a certain angle *θ* from the main axis, the directivity’s lobe rotates by a smaller angle (see Figure 11);-The reverse orientation of the wave relative to the orientation of the directional actuator when certain vibration modes are excited in it (see Figure 11c).

### 3.2. Interfacial Shear Stress and Radial Tangent Displacement Distributions: FE Simulation Results

In order to obtain insights about issues that matter for the wave orientation, the postprocessing results of the interfacial stresses and displacements were analyzed at the modeled excitation frequencies. The normal stresses demonstrated an irregular plane-like noisy character with a small growth when approaching the edges of the footprint, whereas the radial shear stresses amplitudes demonstrated a rather complex, but pronounced spatial distribution corresponding to the natural oscillation modes of the actuator (see Figure 12).

Despite the similarity of these tangent radial stress patterns for each excitation frequency (especially at the frequency 65 kHz), the dynamic response of the excited structure depends on the structural and the damping anisotropy of the material, and also on the panel thickness, which determines the structural stiffness near the actuator location [21]. Two patterns of the amplitude radial tangent displacement at the same frequency, but at the different actuator orientations along (a) and across (b) the main axis of the orthotropic panel (see Figure 13) illustrate this statement. As can be seen, the displacements in the direction of greater structural compliance are significantly higher.

### 3.3. Attenuation of the Lamb Waves Generated by the Differently Oriented Wedge Actuator

The comparison of the two patterns in Figure 13 shows that at *θ* = 90° the displacement amplitude in the Y direction exceeds the displacement in the X direction, which is explained by the lower stiffness of the material in the Y direction. Nevertheless, the directivity diagrams in Figure 11 demonstrate that the lobe’s value along the Y direction is sufficiently less compared to the lobe’s maximum value along the X direction. Obviously, it depends on the attenuation of the waves propagating along the directions, which correspond to the maximum of the wave’s directivity lobe. The directivity diagrams, which are presented in Figure 11, were averaged for two distances from the actuator. Because the most efficient use of each wave can be reached for the directions, corresponding to the directivity lobe (or lobes), it is very important to estimate the attenuation of the waves propagating along the lobe (or lobes) of maximum intensity. For this purpose, the traveling waves amplitudes were calculated for three distances 15 cm, 17.5 cm, and 20 cm from the actuator footprint’s center during simulation of all FE models for the studied system “CFRP panel-wedge actuator”.

These results are presented in Figure 14. Taking into account that the vertical axis of the plots in Figure 14 is presented in logarithmic coordinates, it should be concluded that the Lamb waves amplitude at any angular orientation and the distance from the sound source is exponentially decreasing with the growth of the frequency. The wave also exhibits the highest attenuation, with an increase angle, *θ*, that can be explained by the anisotropic damping of the studied CFRP composite material and consistent with the results of the works [20,21,22,23,24,25,26,27].

All of the abovementioned problems, including a correct determination of the spatial, frequency, and strain type dependent composite material damping, a reasonable choice of the excited acoustic waves, taking into account the possibility of their oriented radiation and the proper choice of the type and the dynamic characteristics of the actuator for this purpose, were not solved till now with the necessary completeness required for the effective practical use of acoustic-based SHM and NDE of modern composite materials and structures. This is due to the fact that these problems are closely related to each other and cannot be solved separately. Our analysis of the previous works, which proposed the analytically formulated and solved models, showed that these approaches lose their effectiveness when trying to describe wave phenomena with the necessary accuracy in highly anisotropic composite structures excited by the devices of complex structure and dynamics behavior. The relatively simple example considered in our article illustrates one of the possible approaches to solving only part of the problem of the rational design of such SHM and NDE systems.

## 4. Conclusions

The research objective of the presented paper was an estimation of the angle-beam wedge actuator ability to generate the spatially oriented acoustic waves in the thin-walled orthotropic CFRP panels with the high anisotropic damping, so that it can be used in SHM and NDE systems. This study was carried out both numerically and experimentally, in a frequency range of 10–100 kHz; these limits were chosen due to a very intensive acoustic-waves attenuation at higher frequencies. The CFRP panel under study always was considered as a one-layered orthotropic composite with the elastic moduli, which previously was determined experimentally. The dispersion analysis, which was based on the semi-analytical partial-waves approach, and for which validity was confirmed experimentally by measuring the wavelength and wave speed at the varied frequency, revealed only two wave types that can be excited in this panel. These were zero-order A0 Lamb wave and SS0 shear horizontally polarized wave. However, the SS0 wavelength in the used frequency range is too long to be used in SHM systems.

In order to formulate the FE model of the overall dynamic system “CFRP panel—wedge actuator” the FE model of this PZT based device was designed in the Comsol Multiphysics 3.5a software by using the actuator sketch and parts mechanical properties provided by manufacturer and the frequency response functions of the contact surface displacement and the consumed electric power, which were studied experimentally. As the results of this study, three eigenfrequencies and natural vibration modes were chosen, at which the actuator supplied the most intensive contact surface displacement amplitude. At these natural frequencies (15, 30, and 65 kHz), all subsequent numerical and experimental studies were performed.

The next study included the wave directivity simulation at the different angles, *θ*, between the actuator’s axis and the main axis of anisotropy of the CFRP panel. A0 Lamb waves directivity was studied both numerically and experimentally, whereas SS0 waves directivity was studied only numerically. Our results demonstrated a very complex dependence of the directivity diagrams on this angle, *θ*; and this dependence is determined by the generated wave’s frequency, by the normal oscillation mode of the actuator and by the anisotropic damping of the CFRP panel. The presented results revealed a better ability to change the waves directivity generated by the angle-beam wedge actuator, compared with the omnidirectional ones, but also the limits of these waves orientation controllability that is due to the high anisotropy of the elastic and damping properties of such composite structures.

The results obtained at the study of the relatively simple example that was considered prove the need to use techniques similar to those presented in this article for an adequate design and a justified estimation of abilities of the acoustic-based SHM systems for the non-destructive testing and evaluation of composite materials and structures.

## Figures and Tables

**Figure 1 materials-13-00907-f001:**
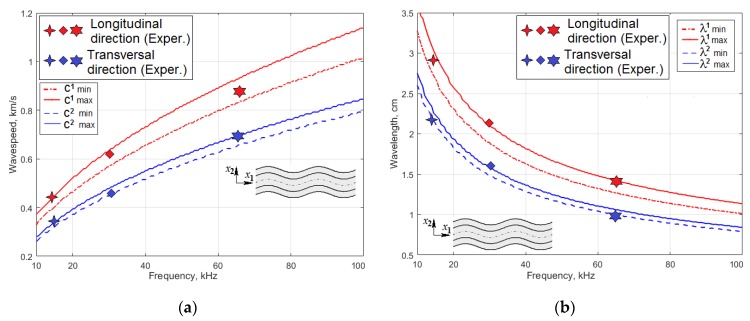
The dispersion curves for the wavespeeds (**a**) and wavelengths (**b**) of the anti-symmetric A0 Lamb waves propagating along (^1^) and across (^2^) the main axis of orthotropic symmetry.

**Figure 2 materials-13-00907-f002:**
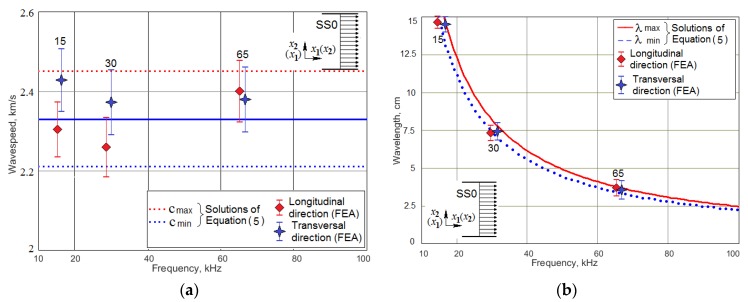
The dependencies of the wavespeeds (**a**) and the wavelengths (**b**) on the frequency for the symmetric horizontally polarized shear waves SS0 propagating along and across the main axis $x_{1}$ of orthotropic symmetry.

**Figure 3 materials-13-00907-f003:**
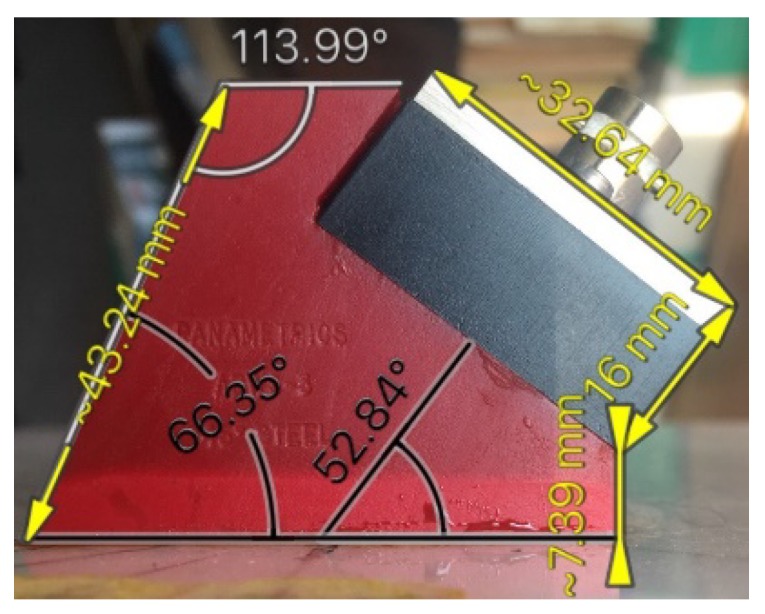
The linear and angular outer dimensions of the used wedge actuator [41].

**Figure 4 materials-13-00907-f004:**
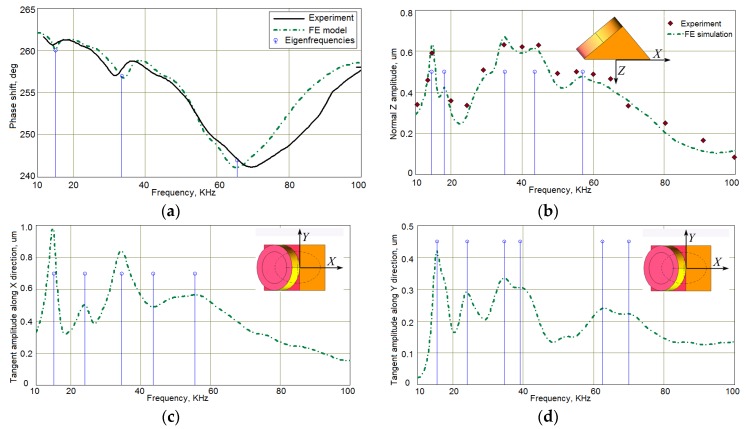
The frequency response functions for the Olympus actuator: (**a**)—phase angle of PZT current; (**b**–**d**)—averaged displacement amplitudes on the contact footprint surface—normal (**b**) and tangential (**c**,**d**).

**Figure 5 materials-13-00907-f005:**
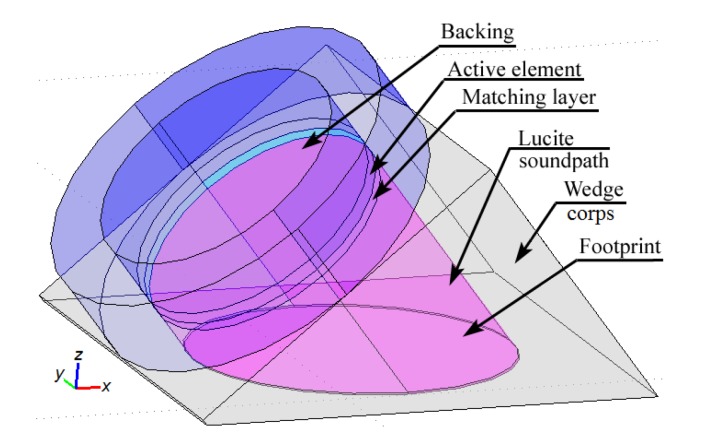
The geometry of wedge actuator’s finite-element (FE) model [41].

**Figure 6 materials-13-00907-f006:**
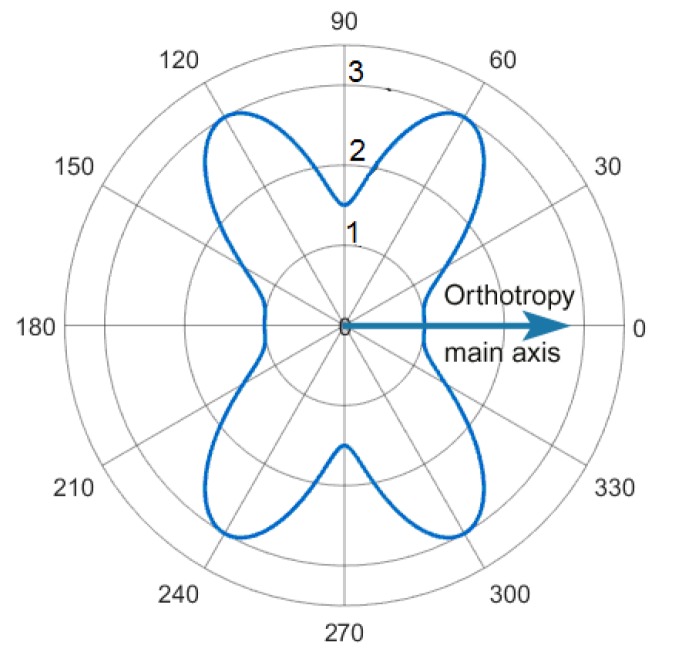
The angular distribution of relative dimensionless attenuation coefficient for the anti-symmetric A0 waves propagating from the center of the orthotropic CFRP panel [20].

**Figure 7 materials-13-00907-f007:**
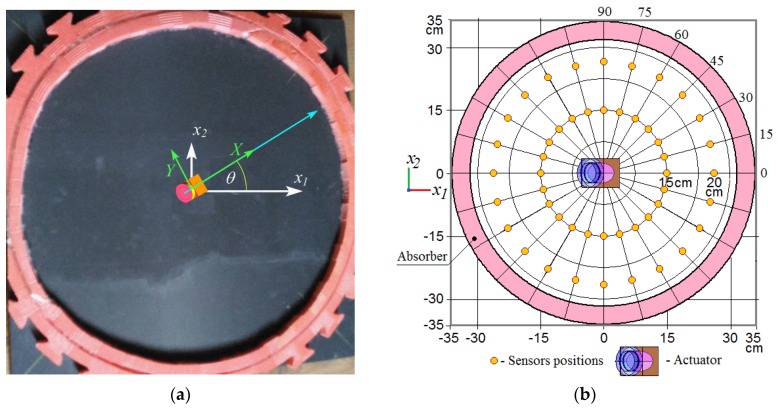
The experimentally studied CFRP panel with schematically shown coordinate systems of orthotropic symmetry $\left(x_{1},x_{2}\right)$ and wedge actuator’s (X,Y) rotated relative to the first one by an angle, *θ* (**a**); the arrangement of sensors for recording the A0 Lamb waves propagation (**b**).

**Figure 8 materials-13-00907-f008:**
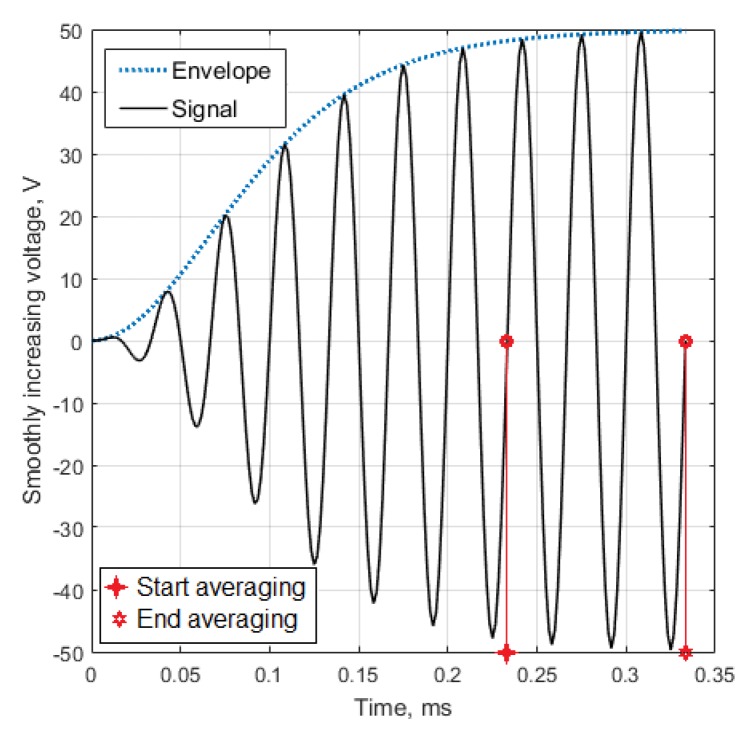
The driving voltage for the PZT actuator and the time interval for the sensor’s signal averaging.

**Figure 9 materials-13-00907-f009:**
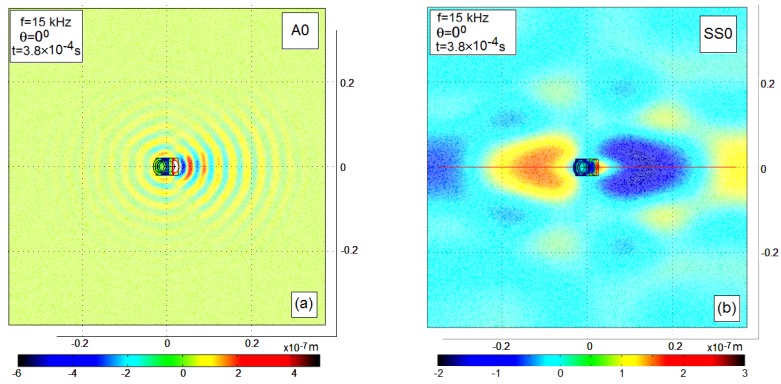
The snapshots of the Lamb A0 (**a**,**c**) and SS0 (**b**,**d**) waves, which propagated in the studied CFRP panel at the frequencies 15 kHz (**a**,**b**) and 30 kHz (**c**,**d**). Color scales show the normal out-of-plane displacements (**a**,**c**) and the radial in-plane displacements (**b**,**d**) at the time instants when these displacements have the amplitude values.

**Figure 10 materials-13-00907-f010:**
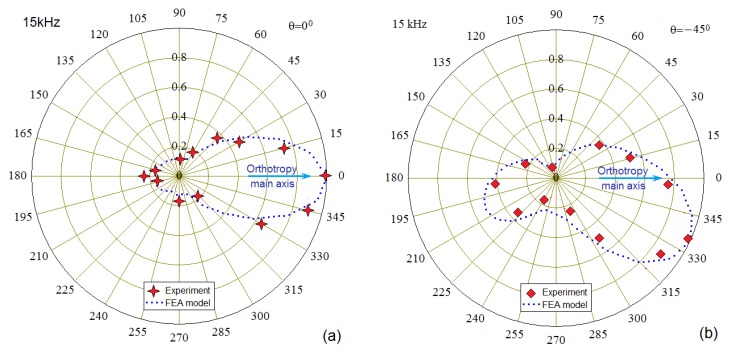
The directivity diagrams of the A0 Lamb waves propagating at the different wedge actuator orientation for the excitation frequencies 15 kHz (**a**,**b**), 30 kHz (**c**,**d**), and 65 kHz (**e**,**f**).

**Figure 11 materials-13-00907-f011:**
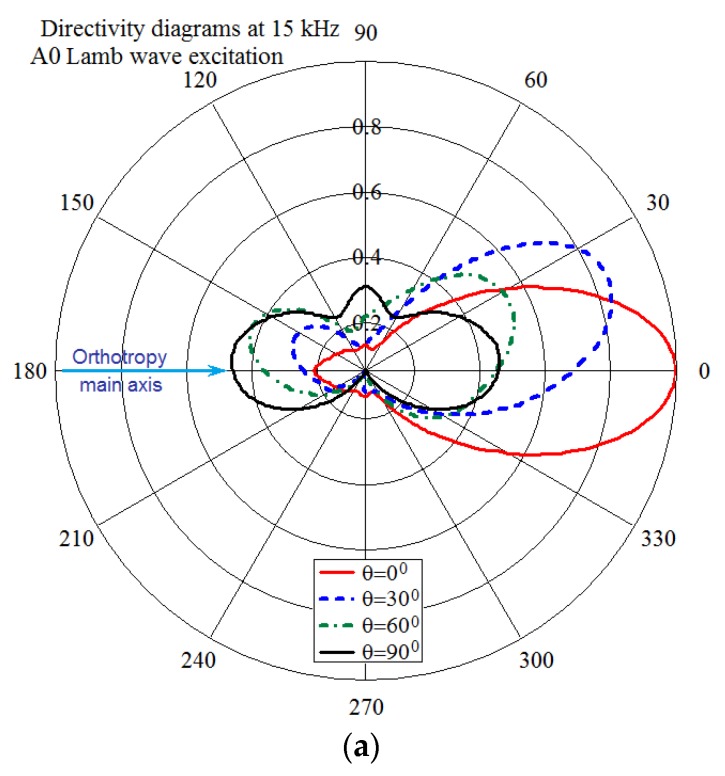
Variation of the A0 Lamb waves directivity with the wedge actuator’s angular orientation relative to the main axis of orthotropic symmetry of the CFRP plate at the excitation frequencies 15 kHz (**a**), 30 kHz (**b**), and 65 kHz (**c**).

**Figure 12 materials-13-00907-f012:**
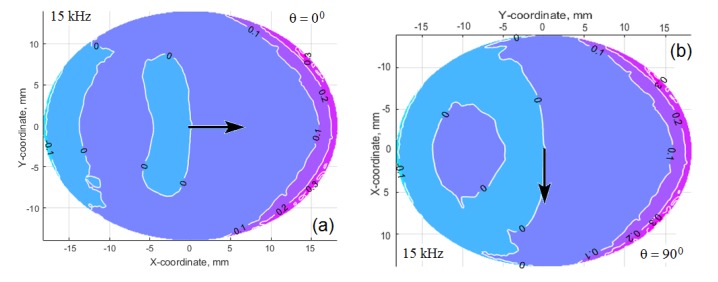
In-plane contact radial stress distributions (amplitude values in MPa) for the waves excitation frequencies 15 kHz (**a**,**b**), 30 kHz (**c**,**d**), and 65 kHz (**e**,**f**) at the actuator orientation along (**a**,**c**,**e**) and across (**b**,**d**,**f**) the main axis of orthotropic panel, which is denoted by the arrows.

**Figure 13 materials-13-00907-f013:**
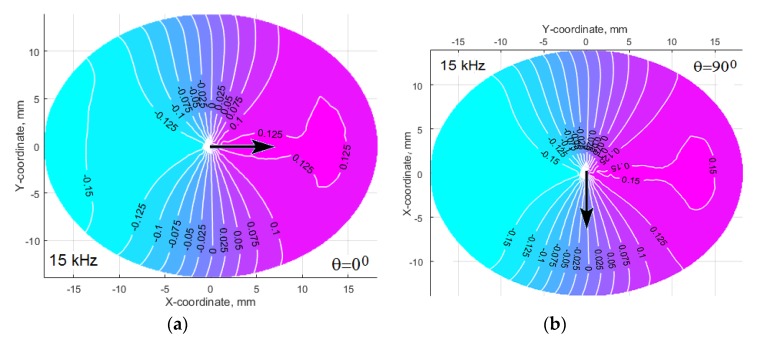
The distribution of the interfacial tangent radial displacements in μm at the excitation frequency 15 kHz. Actuator’s axis is oriented along (**a**) and across (**b**) the main orthotropy axis of the CFRP panel, which is denoted by the arrows.

**Figure 14 materials-13-00907-f014:**
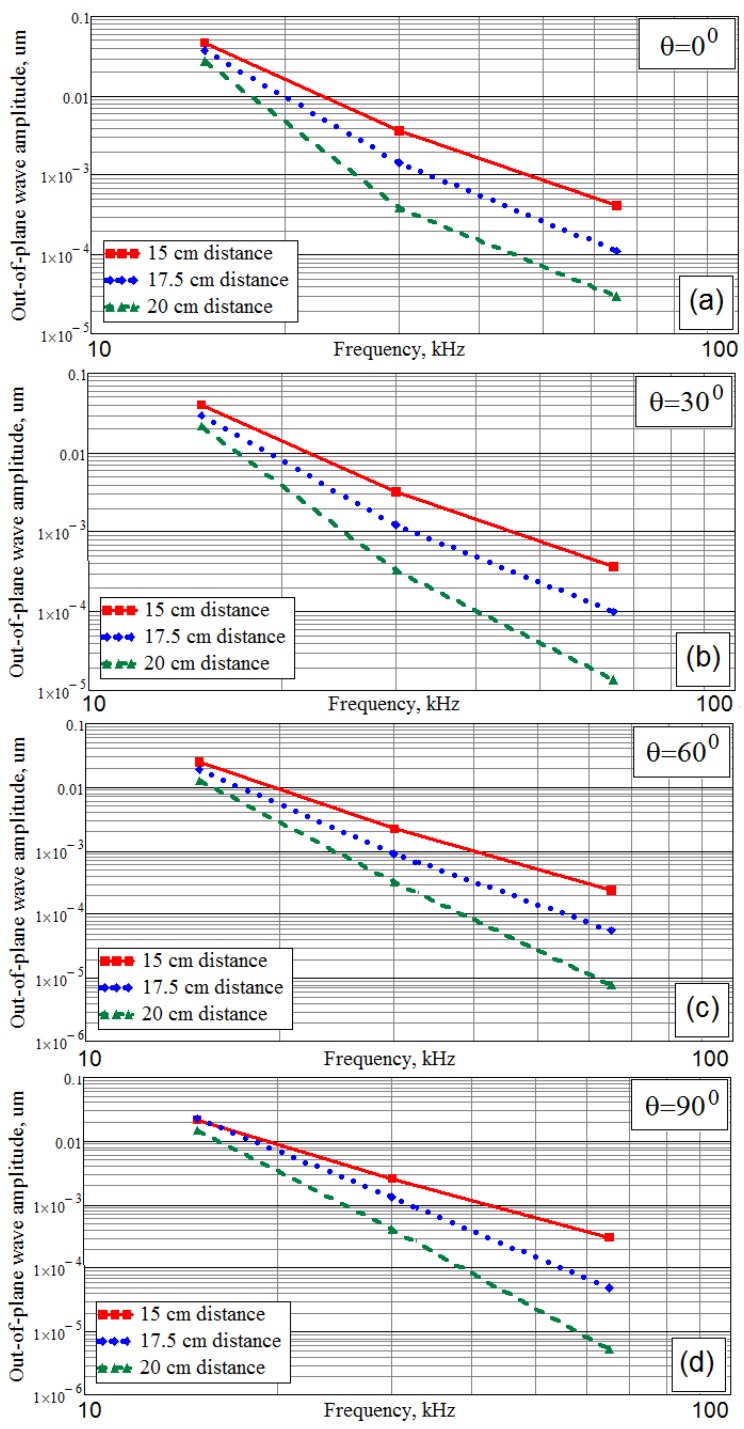
The attenuation of the A0 Lamb waves propagating along the main lobes of the directivity diagrams at the different wedge actuator orientation: along the main orthotropic axes of the material (**a**), and rotated 30, 60 and 90 degrees (**b**,**c**,**d**, respectively) and the waves excitation frequencies.

**Table 1 materials-13-00907-t001:** The engineering elastic constants of CFRP panel ^1^.

Young’s Moduli, GPa			Shear Moduli, GPa			Poisson’s Ratios		
*E* _1_	*E* _2_	*E* _3_	*G* _12_	*G* _23_	*G* _31_	*ν* _12_	*ν* _23_	*ν* _13_
63 ± 12	22.5 ± 2.5	5 ± 1.5	9.6 ± 1.2	7.3 ± 0.6	7.8 ± 0.6	0.46 ± 0.06	0.32 ± 0.08	0.6 ± 0.1

^1^ The bounds of the confidence intervals for the elastic moduli are determined by using measuring data.

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
