# Peer review of "On the Directivity of Lamb Waves Generated by Wedge PZT Actuator in Thin CFRP Panel"

_materials, 2020, doi:10.3390/ma13040907_

Round 1
Reviewer 1 Report
This work seeks wave propagation behavior in FRPs. However, this has been available in many previous studies and Disperse software allows one to obtain the same. While some results appear new, it is difficult to believe what are presented because the lack of detailed description of models used, experimental methods, parameters such as specifics of GFRP/CFRP used, sensor model, etc. prevents assessment of the work given. No validation of models is shown. Further, the choice of frequency 10-100 kHz was made for low attenuation and elastic moduli have no imaginary components, wave attenuation is analyzed (fig 14). This can only be from geometrical spreading, not from inherent absorption. In fact, no citation appears from Castaing, which indicated that CFRP attenuation is much higher than isotropic PMMA or epoxy (using complex elastic constants) contrary to what the AUs assumed. Strange results are also given. E.g. 1: Eq 4 a and b are for orthotropic solids, but includes Lame constants for velocity that only are valid for isotropic media. E.g. 2: Fig 2 shows dispersive Ao despite the fact that the para above (L 326-334) says only “changeless” speed (constant speed?) was observed.
I recommend rejection and no re-submission.
Author Response
Response to reviewer 1 (corrections in the paper text and our answers made in red)
Dear reviewer! We will try to answer some of your questions so that you may change your mind. We are also confident that answers to your questions and taking into account your advice will help improve the quality of the materials presented.
This work seeks wave propagation behavior in FRPs. However, this has been available in many previous studies and Disperse software allows one to obtain the same.
Naturally, questions adjoining the problems of our article have been repeatedly studied, and we refer in the text to many of these works. In particular, fundamental works are known that studied the features of the propagation of acoustic waves in orthotropic laminates by using the omnidirectional (circular and annular) actuators. In a number of works of recent years, the possibilities of wedge actuators exciting waves in thick-walled structures made of isotropic materials (metals) are also studied. These studies use the finite element methods because complexity of the angle-beam wedge actuators design doesn't allow to apply any analytical approach. But no studies that investigated the possibility of generation the wave with controllable directivity in highly anisotropic plate (which are composite structures most often) by using the actuators able to generate the spatially oriented acoustic waves. This circumstance justifies the choice of the object of our study.
While some results appear new, it is difficult to believe what are presented because the lack of detailed description of models used, experimental methods, parameters such as specifics of GFRP/CFRP used, sensor model, etc. prevents assessment of the work given.
Unfortunately, due to limiting volume of the paper we eliminated the detailed description of the components of our study mentioned by reviewer. We references to our previous works [20, 21, 41], where these components are described in detail. But completely agreeing with the opinion of the reviewer, we took into account the majority of these comments in the revised text. We inform the readers the type and properties of used CFRP unidirectional lamina (lines 274-276), the standards used at the mechanical testing of the laminated CFRP panel (lines 279-282), and sensors used in the experimental part of our study (see line 396). Also, we explain in the revised text, why we do not give a description of the model equations of the mechanics of anisotropic materials and piezoelectricity, which we use in finite element calculations (see lines 352-354).
Further, the choice of frequency 10-100 kHz was made for low attenuation and elastic moduli have no imaginary components, wave attenuation is analyzed (fig 14).
This frequency range was chosen due to high structural damping of CFRP material that is cause of intensive excited waves attenuation. The material damping of all parts of the modeled system including not only CFRP panel's material, but also the parts of the actuator, were taken into account in our FE model using the Rayleigh damping coefficients, which have been determined experimentally as described in detail in our paper [19 - Piezoelectric Based Lamb Waves Generation and Propagation in Orthotropic CFRP Plates: I. Influence of Material Damping. Mater Sci Forum 2019, Vol. 962, pp. 218-226.] (see lines 369-378).
This can only be from geometrical spreading, not from inherent absorption. In fact, no citation appears from Castaing, which indicated that CFRP attenuation is much higher than isotropic PMMA or epoxy (using complex elastic constants) contrary to what the AUs assumed.
Dear reviewer! We didn't cited the papers of Dr. P. Castaing because these problems were subordinate in his publications. In the above article, we cited the following works devoted exclusively to the damping of reinforced polymeric composites and the attenuation of acoustic waves in them. In particular, we used the experimental technique for separate determination of the contributions of geometric and material attenuation. So, waves attenuation data and appropriate empiric model take into account namely the effect of the material damping in the waves attenuation. This part of the study is not presented in the peer-reviewed article, because it was previously published by us.
Gresil M., Giurgiutiu V. Prediction of attenuated guided waves propagation in carbon fiber composites using Rayleigh damping model // Journal of Intelligent Material Systems and Structures,2014, Vol. 26(16) pp.19 Ramadas C. Three-dimensional modeling of Lamb wave attenuation due to material and geometry in composite laminates // Journal of Reinforced Plastics and Composites 2014 V.33: pp.824-835 Crane R.M. Vibration Damping Response of Composite Materials // David Taylor Research Center Report (1991), 302 p. Adams, R.D. and Bacon, D.G.C., "Effect of Fibre Orientation and Laminate Geometry on the Dynamic Properties of CFRP," Journal of Composite Materials, Vol. 7, Oct. 1973, pp. 402-428 Saravanos D.A., Chamis C.C. Computational Simulation of Damping in Composite Structures // NASA Tech.Report (1989), 27 p. Maheri M.R. and Adams R.D., Modal Vibration Damping of Anisotropic FRP Laminates Using the Rayleigh-Ritz Energy Minimization Scheme, Journal of Sound and Vibration, 259(1) (2003) 17–29 Lonne S. et al. MODELING OF ULTRASONIC ATTENUATION IN UNIDIRECTIONAL FIBER REINFORCED COMPOSITES COMBINING MULTIPLE-SCATTERING AND VISCOELASTIC LOSSES // Review of Quantitative Nondestructive Evaluation, 2004, Vol. 23, pp. 875-882 Gao Y., Li Y., Zhang H., He X. Modeling of the Damping Properties of Unidirectional Carbon Fibre Composites // Polymers & Polymer Composites, Vol. 19, Nos. 2 & 3, 2011, pp. 119-122 Ono K., Galego A. Attenuation of Lamb Waves in CFRP Plates // J. Acoustic Emission, 30 (2012), pp. 109-123 Mei H., Giurgiutiu V. Effect of structural damping on the tuning between piezoelectric wafer active sensors and Lamb waves // Journal of Intelligent Material Systems and Structures, 2018, Vol. 29(10), pp.2177-2191
Fig 2 shows dispersive Ao despite the fact that the para above (L 326-334) says only “changeless” speed (constant speed?) was observed.
The revised Figure 2 shows the dependencies of the wave wavespeed and wavelengths on the frequency for the horizontally polarized shear waves SS0 (not A0 wave!). The behavior of these curves are clear: at the constant wavespeed variation of the frequency lead to the change of wavelength: λ=c/f (simple hyperbola). To eliminate a misunderstanding, we changed caption for Figure 2. Thank you for your remark.
Strange results are also given. E.g. 1: Eq 4 a and b are for orthotropic solids, but includes Lame constants for velocity that only are valid for isotropic media. E.g. 2: Fig 2 shows dispersive Ao despite the fact that the para above (L 326-334) says only “changeless” speed (constant speed?) was observed.
Thanks to your comment, the text you marked is completely rewritten (lines 292-309).
Dear reviewer! We bring you our deepest gratitude for our unforgivable mistake you have identified. I did not notice that my young colleague inserted material related to isotropic material into the article.
Reviewer 2 Report
Lines 77-78 -> the paragraph starting with line 78 appears to be unconnected to the previous paragraph, causing the loosing of the text sense. I suggest connecting the two paragraphs in a more refined way.
Lines 101-106 -> again, the second sentence (lines 104-106) appears to be unconnected to the previous sentence. I suggest connecting the two sentences in a more refined way.
Line 147-148 -> I suggest clarifying how the “understanding of these characteristics” is beneficial for SHM/NDE applications.
Line 159-160 -> the meaning of the sentence is not clear, please clarify it.
Line 161-163 -> it is not clear the motivation for making a comparison between the frequency ranges adopted by other authors and what advantages this brings to the work under investigation. Please clarify this point.
Line 165 -> it seems that a reference is missing.
Line 224 -> please, pay attention to the references.
Line 320 -> please, pay attention to the rho symbol.
Figures 1 and 2 -> It is not clear from where the experimental frequencies used come from and their values.
Lines 355-356 -> it is not clear if the normal displacement is experimentally measured during the work or in the previous publication (ref [41]). Moreover, it is not clear to what “the calibrated sensors” refers to. Please, clarify these points.
Figure 4 -> please use the same line style for the same quantity; using the dashed-dot style for FE model results in Fig.4a and the same style for the experimental results in Fig.4b, generates confusion. Moreover, the legends in Fig.4c and d are missing.
Line 383 -> what C0 and C1 are? Their meaning is not explained.
Line 408 -> it is not clear how the 15, 30 and 65 kHz frequencies are chosen, especially in relation to the results obtained in sub-chapters 2.1 and 2.2. I suggest clarifying this point.
Figure 9 -> in the caption, “SS0” instead of “SSH”.
Line 428 -> Figure 10 instead of Figure 9.
Line 431 -> Figure 11 instead of Figure 10.
Lines 478-482 -> it is not clear the x1 and x2 directions, as they are not reported in the figures mentioned in the text.
Line 488 -> it is not clear which the “the three distances” are.
Line 514 -> I suggest reporting the frequency range used in the work.
In general:
The introduction is very long and include a lot of technical details that don’t allow an easy understanding of the statements, for the authors work definition. I suggest a general introduction rearrangement, with a dedicated chapter for the theoretical discussion of the phenomena, affecting the ultrasonic waves propagation in anisotropic materials. Sometimes, the sentences are long and full of information. I suggest using shorter sentences, focused on a single statement/group of statements. General language and text revision.Author Response
Response to reviewer 2 (corrections in the paper text and our answers made in blue)
Lines 77-78 -> the paragraph starting with line 78 appears to be unconnected to the previous paragraph, causing the loosing of the text sense. I suggest connecting the two paragraphs in a more refined way.
Another important property characterizing the ability of acoustic waves to interact with possible defects is the range of their propagation, or, their opposite property is attenuation. (lines 73-74)
Lines 101-106 -> again, the second sentence (lines 104-106) appears to be unconnected to the previous sentence. I suggest connecting the two sentences in a more refined way.
But the dissipative properties of the fiber reinforced composites depend on the many factors, and these dependencies should be taken into account at the estimation of the range of wave advancement to supply the desired parameters of SHM system. These dependencies have been studied by many researchers.
In the early study [23] it was established that fibers material usually has extremely low damping capacities and so can be considered not to contribute to the damping of the composite, whereas a large proportion of the dissipated energy must be in the matrix. The work [24] reported about many experiments implemented with different kinds of CFRP and GFRP laminas demonstrated a significant growth of the damping parameters (up to ten times and more) with the change of a cyclic strain orientation relative to the fibers direction. (lines 95-104)
Line 147-148 -> I suggest clarifying how the “understanding of these characteristics” is beneficial for SHM/NDE applications.
The authors of the paper [8] concluded that knowledge of the attenuation and directivity of the waves, depending on the anisotropy of the elastic and damping properties of the material for specific types of waves and their excitation frequencies will benefit future acoustic based SHM and NDE applications in composite materials design and technology. (lines 134-138)
Line 159-160 -> the meaning of the sentence is not clear, please clarify it.
The experimental technique [9], which allows to determine the angular distribution of the attenuation coefficient, was used in our previous work [19]. These results we use in the present paper. (lines 146-148)
Line 161-163 -> it is not clear the motivation for making a comparison between the frequency ranges adopted by other authors and what advantages this brings to the work under investigation. Please clarify this point.
Despite the fact that many theoretical papers report frequencies of the used waves of the order of several MHz, articles presenting the results of neatly performed experiments with the polymeric composites inform of much lower frequencies. (lines 149-151)
Line 165 -> it seems that a reference is missing.
The later studies [1, 9-11] using modern advanced equipment were able to operate at higher frequencies. (lines 152-153)
Line 224 -> please, pay attention to the references.
Some modernizations of the wedge actuators allowing to eliminate the reflections at the wedge-load interface and multiple reflections inside the wedge were described in [33-36]. (lines 211-212)
Line 320 -> please, pay attention to the rho symbol.
This part of the paper was revised, and "rho" symbol spelling was corrected (line 297)
Figures 1 and 2 -> It is not clear from where the experimental frequencies used come from and their values.
The results in the form` of the dispersion curves for two values of angle are shown in Figures 1, a,b together with the experimentally measured values at the frequencies, which correspond to the natural oscillation modes of actuator (see below). (line 309-312)
The dependencies of the SS0 wave speed and wavelength on the frequency are shown in Figures 2, a,b together with the results of the FE simulation at the actuator's first eigenfrequencies. (line 317-319)
Lines 355-356 -> it is not clear if the normal displacement is experimentally measured during the work or in the previous publication (ref [41]). Moreover, it is not clear to what “the calibrated sensors” refers to. Please, clarify these points.
Only averaged normal displacements amplitude of the actuator's contact surface were measured experimentally by using the VibroGo single-point vibrometer (Polytec Co.®), which were sensitive to the normal acceleration at the frequencies up to 100 kHz. (line 343-346)
Figure 4 -> please use the same line style for the same quantity; using the dashed-dot style for FE model results in Fig.4a and the same style for the experimental results in Fig.4b, generates confusion. Moreover, the legends in Fig.4c and d are missing.
We corrected an erroneous lines description in Fig. 4,b and explained the Figs. 4, c,d in text (line 350-351).
We want to explain the reason for the different appearance of the experimental data in Figures 4, a, b. The frequency response of the consumed electric current in Figure 4, a was obtained at the output of a device performing continuous scanning in frequency, while the measurement of the amplitudes of normal displacements (Fig. 4, b) was performed as a series of measurements on a discrete set of frequencies.
Line 383 -> what C0 and C1 are? Their meaning is not explained.
The FE model of the studied system provided this attenuation inside the circle of radius 27.5 cm by the Rayleigh damping coefficients, which depend on the orientation in a polar coordinate system. But outside this circle the damping coefficients increased quadratically while maintaining C0 and C1 continuities (of the damping coefficients distribution and their spatial derivatives, respectively) at the boundary of the circle. (lines 384-386)
Line 408 -> it is not clear how the 15, 30 and 65 kHz frequencies are chosen, especially in relation to the results obtained in sub-chapters 2.1 and 2.2. I suggest clarifying this point.
The directivity of the waves generated by the wedge actuator has been studied at the three frequencies: 15, 30 and 65 kHz, which were chosen due to the highest intensity of the waves excited at these frequencies, which are the eigenfrequencies of the actuator's structure. (line 411-413)
Figure 9 -> in the caption, “SS0” instead of “SSH”.
Thank you VM! Figure 9 caption was corrected.
Line 428 -> Figure 10 instead of Figure 9.
Sorry! Reference to the Figure was corrected.
Line 431 -> Figure 11 instead of Figure 10.
The correction you recommended is done. Thank you.
Lines 478-482 -> it is not clear the x1 and x2 directions, as they are not reported in the figures mentioned in the text.
The designations in the explanatory text are corrected in accordance with the Figures 13, a,b.
Line 488 -> it is not clear which the “the three distances” are.
We clarified the meaning and slightly reformatted this sentence (line 477-480):
For this purpose, the traveling waves amplitudes were calculated for the three distances 15, 17.5 and 20 cm from the actuator footprint's center during simulation of all FE models for the studied system "CFRP panel - wedge actuator".
Line 514 -> I suggest reporting the frequency range used in the work.
Yes, of course. Your suggestion was used and inserted in text (line 514)
In general:
The introduction is very long and include a lot of technical details that don’t allow an easy understanding of the statements, for the authors work definition. I suggest a general introduction rearrangement, with a dedicated chapter for the theoretical discussion of the phenomena, affecting the ultrasonic waves propagation in anisotropic materials. Sometimes, the sentences are long and full of information. I suggest using shorter sentences, focused on a single statement/group of statements. General language and text revision.
Dear reviewer! We rearranged and shortened the Introduction of the paper according to your recommendations. Long expressions, errors in the text, clerical errors and unclear expressions were generally excluded. Thank you very much for your detailed and very useful analysis of our paper.
Reviewer 3 Report
The purpose of this paper was to determine the capability of actuators to provide the controlled generation of acoustic wave of desirable type with the necessary wavelength, propagation distance and directivity. Paper requires some adjustments. Authors should answer the following questions and make changes in the text.
Introduction
The review of reference sources was carried out correctly. Formulas (1) (2) (3) are not necessary in the introduction. Authors may include them in the Materials and Methods chapter.
Materials and Methods
The dispersion curves given on the Fig. 1 and 2 with the mathematical formula and R2 coefficient should be described.
Results and Discussion
The linear curves given on the Fig. 14 with the mathematical formula and R2 coefficient should be described.
Conclusions
The first sentences “… In this paper we presented some experimental and numerical modeling investigations results of the directed guided-wave excitation by angle-beam wedge piezoelectric actuator in a thin-walled multi-layered composite panel with orthotropic symmetry of the material. Our investigation included the dispersion analysis of the waves that can be excited in the studied panel at the limited frequency range, whose upper limit is caused by the frequency dependent waves attenuation. …” are not a conclusion. Authors should remove them. The Authors should add more conclusions based on the analysis.
References
The selection of the references was made correctly.
Paper can be published after minor changes.
Author Response
Response to reviewer 3 (corrections in the paper text and our answers made in green)
Formulas (1) (2) (3) are not necessary in the introduction. Authors may include them in the Materials and Methods chapter.
These formulas were excluded from the Introduction. Thank you very much for your suggestion
The dispersion curves given on the Fig. 1 and 2 with the mathematical formula and R2 coefficient should be described.
The text, which relates to the Figs. 1, 2 was fully reorganized taking into account the suggestions of reviewers 1 and 3. We consider the changes made very important and useful. Thank you for your advice.
The linear curves given on the Fig. 14 with the mathematical formula and R2 coefficient should be described.
The part of the article text, which describes the Figs.14, was fully reorganized according to the suggestions of the reviewer 2 and 3 to clarify the sense of its description. (line 468-492)
Conclusions
The first sentences “… In this paper we presented some experimental and numerical modeling investigations results of the directed guided-wave excitation by angle-beam wedge piezoelectric actuator in a thin-walled multi-layered composite panel with orthotropic symmetry of the material. Our investigation included the dispersion analysis of the waves that can be excited in the studied panel at the limited frequency range, whose upper limit is caused by the frequency dependent waves attenuation. …” are not a conclusion. Authors should remove them. The Authors should add more conclusions based on the analysis.
The text of Conclusion was fully rewritten according to your recommendations. (lines 511-544).
Dear reviewer! Your advices have significantly improved the quality of the text of the article and its information content.
Reviewer 4 Report
The review concerns the article titled "On the Directivity of Lamb Waves Generated by 2 Wedge PZT Actuator in Thin CFRP Panel" by Sergey Shevtsov, Valery Chebanenko, Maria Shevtsova, Shun-Hsyung Chang, 4 Evgenia Kirillova and Evgeny Rozhkov.
In my opinion, the presented article is interesting and deserves attention.
The article presents very important and new issues in the aspect of monitoring the condition of composite structures. The subject of the article is current.
The article is thematically consistent with the objectives of the Materials.
The article is well written. I do not submit comments to the article.
Author Response
Response to reviewer 4
Dear reviewer! We are very grateful to you for your attention to our work and for its good assessment, despite the presence of a significant number of mistakes and inaccuracies in it. We hope that we have managed to eliminate them.
Reviewer 5 Report
Skimming quickly through the pages, the manuscript looks well organized but a more careful reading shows some critical issues: starting from the abstract, it is possible to find a lack of indefinite and definite articles, continuous changes between present and past tenses (that continue throughout the text). There is not a progressive bibliography numbering in the text. Why does the number 17 anticipate the number 14? Similar cases are present in other pages. At line 224, the citation is repeated twice: [33, 33]. In the page 7, the authors should explain better how they obtained the properties for the equivalent orthotropic plate. It is a crucial step for the numerical results. For experience, it is not possible to guarantee the right behaviour at each direction. Furthermore, there is a strong dependence both on the considered mode propagating in the structure and the frequency. At ultrasound frequencies, the shear becomes dominant. I think you should calibrate the model properly according to the propagation characteristics and not by static test as you assert at line 306. Finally, it would be better to provide a description about the modelling by finite element, being the manuscript based even on it.
Author Response
Response to reviewer 5 (corrections in the paper text and our answers made in orange)
Skimming quickly through the pages, the manuscript looks well organized but a more careful reading shows some critical issues: starting from the abstract, it is possible to find a lack of indefinite and definite articles, continuous changes between present and past tenses (that continue throughout the text). There is not a progressive bibliography numbering in the text. Why does the number 17 anticipate the number 14? Similar cases are present in other pages. At line 224, the citation is repeated twice: [33, 33].
Thank you very much for your accurate observations. References list was corrected. All noted errors of this kind are resolved.
In the page 7, the authors should explain better how they obtained the properties for the equivalent orthotropic plate. It is a crucial step for the numerical results. For experience, it is not possible to guarantee the right behaviour at each direction.
We described the methods used for elastic and damping properties in more detail, also referring to the Standards and borrowed methods from the studied works. (lines 279-283, 369-378)
Furthermore, there is a strong dependence both on the considered mode propagating in the structure and the frequency. At ultrasound frequencies, the shear becomes dominant. I think you should calibrate the model properly according to the propagation characteristics and not by static test as you assert at line 306.
Yes, of course. In the revised version we described shortly, how the damping properties dependence on the frequency was determined by using experimental study of the wave attenuation. (lines 369-378)
Finally, it would be better to provide a description about the modelling by finite element, being the manuscript based even on it.
We have provided a brief description of the FE technology used (lines 351-355)
Dear reviewer! We express our sincere gratitude to you for a thorough analysis of our work, which allowed us to identify many of its shortcomings and to eliminate them.
Round 2
Reviewer 1 Report
All of my previous issues have been addressed. It is recommended for publication.
Author Response
Dear reviewer! We tried to fulfill all your recommendations in the revised text of the article. Thank you very much for your valuable help.
Reviewer 5 Report
The manuscript was written in third person but the new added text lines are formulated in first plural person (e.g. lines 148, 247, 258, 280, 281, 301, 414 etc.). I suggest to adopt the third person alone.
However, the authors failed to address the reviewer’s comments.
Why did the authors perform material tests according to D-2344 (Test Method for Short-Beam Strength) if just the elastic properties are needed?
Moreover, it is not clear the matching between static tests and dynamic behavior. The authors should justify the simplifications made to define equivalent orthotropic plate also through examples from literature and the reasons why simplified modelling approaches [1-4], that should be mentioned as state of the art for the modeling, are neglected.
[1] L. Maio, V. Memmolo, F. Ricci, N. D. Boffa, E. Monaco, R. Pecora, Ultrasonic wave propagation in composite laminates by numerical simulation, Composite Structures, Volume 121, March 2015, Pages 64-74.
[2] S. Shoja, V. Berbyuk, A. Boström, Delamination detection in composite laminates using low frequency guided waves: Numerical simulations, Composite Structures, Volume 203, 1 November 2018, Pages 826-834.
[3] L. Maio, V. Memmolo, F. Ricci, N. D. Boffa, E. Monaco, Investigation on fundamental modes of guided waves propagating in symmetric and nonsymmetric composite laminates, Proceedings of the Institution of Mechanical Engineers, Part C: Journal of Mechanical Engineering Science, Volume 231, Issue 16, Pages 2988-3000.
[4] S. Shoja, A. Boström, V. Berbyuk, Application of low frequency guided waves to delamination detection in large composite structures: a numerical study, Conference: 18th ECCMAt: Athens, Greece, 2018.
Author Response
Dear reviewer! Please, accept our responses and explanations. These are marked by brown here and in the corrections you suggested
The manuscript was written in third person but the new added text lines are formulated in first plural person (e.g. lines 148, 247, 258, 280, 281, 301, 414 etc.). I suggest to adopt the third person alone.
Thank you for your comments. Your suggestion was applied everywhere in the paper text.
However, the authors failed to address the reviewer’s comments.
Why did the authors perform material tests according to D-2344 (Test Method for Short-Beam Strength) if just the elastic properties are needed?
Dear reviewer! We refer to the standard Ð’-2344 because the loading scheme and the sample geometry were made according to this standard. But the calculation of the interlaminar shear module according to the results of loading the sample in the elastic region was carried out according to the method described in our work [21] - Mechanical Testing of Polymeric Composites for Aircraft Applications: Standards, Requirements and Limitations (https://link.springer.com/chapter/10.1007/978-3-319-03749-3_17)
Moreover, it is not clear the matching between static tests and dynamic behavior.
Dear reviewer! Unfortunately, there is no reliable dynamic testing method for determining the full set (9) of otrhotropic elastic moduli. The results of the static test were confirmed for the longitudinal elastic module using DMA testing (cantilever bending) at frequencies up to 200 Hz, but it is very low frequency. Our assumption about the possibility of using static modules at relatively low frequencies is confirmed by the fact that the discrepancies between calculated and measured wavelengths in our experiments were 10-15%, no more, in the entire studied frequency range. This is a difficult problem. We agree.
The authors should justify the simplifications made to define equivalent orthotropic plate also through examples from literature and the reasons why simplified modelling approaches [1-4], that should be mentioned as state of the art for the modeling, are neglected.
[1] L. Maio, V. Memmolo, F. Ricci, N. D. Boffa, E. Monaco, R. Pecora, Ultrasonic wave propagation in composite laminates by numerical simulation, Composite Structures, Volume 121, March 2015, Pages 64-74.
[2] S. Shoja, V. Berbyuk, A. Boström, Delamination detection in composite laminates using low frequency guided waves: Numerical simulations, Composite Structures, Volume 203, 1 November 2018, Pages 826-834.
[3] L. Maio, V. Memmolo, F. Ricci, N. D. Boffa, E. Monaco, Investigation on fundamental modes of guided waves propagating in symmetric and nonsymmetric composite laminates, Proceedings of the Institution of Mechanical Engineers, Part C: Journal of Mechanical Engineering Science, Volume 231, Issue 16, Pages 2988-3000.
[4] S. Shoja, A. Boström, V. Berbyuk, Application of low frequency guided waves to delamination detection in large composite structures: a numerical study, Conference: 18th ECCMAt: Athens, Greece, 2018.
Dear reviewer! In the paper text we refer, in particular, to the following works:
[16] - Raghavan, A. and Cesnik, C.E.S. Modeling of Guided-Wave Excitation by Finite-dimensional Piezoelectric Transducers in Composite Plates. Proceedings of the 48th AIAA/ASME/ASCE/AHS/ASC Structures, Structural Dynamics, and Materials Conference, (Honolulu, Hawaii, 23 - 26 April 2007; American Institute of Aeronautics and Astronautics), 2007; 15 p., which uses a semi-analytical technique based on the assembling of the laminate global matrix from the individual layer matrices, and on the next 2-D spatial Fourier transform to obtain the problem solution.
[10] Gresil, M., Giurgiutiu, V. Prediction of Attenuated Guided Waves Propagation in Carbon Fiber Composites Using Rayleigh Damping Model. J Intel Mat Syst Str 2015. Vol. 26(16), pp. 2151-2169, which uses the effective engineering constants of CFRP to calculate the dispersion relations.
[29] Ostiguy, P.-C., Quaegebeur, N ., Bilodeau, M. and Masson, P. Semi-Analytical Modelling of Guided Waves Generation on Composite Structures Using Circular Piezoceramics. Proc. SPIE 2015, Vol. 9438, Health Monitoring of Structural and Biological Systems 2015, 14 p., which solves the dispersion problem using two aforementioned approaches: Cristoffel equation with the ortotropic elasticity matrix and an approach based on on the assembling of the laminate global matrix.
Thus, both approaches are eligible to be used. We explain our choice of method with the following arguments.
Let's name the proposed approach as LA (assembly of layers), and used in our article - EM (Engineering Modules).
- LA approach does not and cannot, in principle, take into account the influence of the interface between the intersecting layers. This is a main reason why engineering moduli, calculated on the base of the classical lamination theory, always differ from the experimentally measured by 30-40% and more.
- In principle, the wave motions of each layer cannot be identified experimentally, while the wave motions of the laminated composite are always experimentally studied as a homogenous structure with the omitted fact that it consists of many layers.
- At relatively low frequencies, when the wavelength is 1-2 orders of magnitude greater than the layer thickness, especially when the number of layers is large enough, the mesostructure of each individual layer does not affect the effective properties of the plate, which can be considered uniform for such a wave.
- Effective elastic moduli of the multilayered laminate can be determined experimentally with sufficiently high accuracy, automatically ensuring that the influence of the interlayer interface is taken into account.
- Numerical stability of the LA method, which takes into account a behavior of each individual layer, is very poor, and it deteriorates with each increase in the number of layers.
Dear reviewer! Thank you very much for your comments, questions and critical remarks.